# Serological Evidence of Q Fever among Dairy Cattle and Buffalo Populations in the Campania Region, Italy

**DOI:** 10.3390/pathogens11080901

**Published:** 2022-08-10

**Authors:** Gianmarco Ferrara, Barbara Colitti, Ugo Pagnini, Danila D’Angelo, Giuseppe Iovane, Sergio Rosati, Serena Montagnaro

**Affiliations:** 1Department of Veterinary Medicine and Animal Productions, University of Naples, “Federico II”, Via Delpino 1, 80137 Naples, Italy; 2Department of Veterinary Science, University of Turin, Largo Paolo Braccini 2, 10095 Grugliasco, Italy

**Keywords:** *Coxiellosis*, Q fever, serosurvey

## Abstract

Due to its economic impact on livestock and its zoonotic effect, Q fever is a public and animal health problem. Information on this infection in Italy is presently supported by reports of reproductive problems in livestock farms and is, therefore, insufficient to properly understand the impact of the disease. This study aimed to describe for the first time the seroprevalence of Q fever in dairy cows and water buffalos in the Campania region (Southern Italy). A total of 424 dairy cattle and 214 water buffalo were tested using a commercial indirect ELISA kit. An overall seroprevalence of 11.7% confirmed the wide distribution of *C. burnetii* in this region. Several factors were positively associated with higher seroprevalence, such as species (higher in cattle than in water buffalo), age, and coexistence with other ruminant species. The final model of logistic regression included only age (older) and species (cattle), which were positively associated with the presence of Q fever antibodies. Our findings support the widespread presence of *Coxiella burnettii* in Campania and show a seroprevalence similar to that observed in previous studies in other Italian regions and European countries. Since human cases are typically linked to contact with infected ruminants, there is a need to improve surveillance for this infection.

## 1. Introduction

The Q fever causative agent, *Coxiella burnetii*, is a gram-negative, intracellular bacterium of the *Legionellales* order. *Coxiellosis* has a broad host range, including a large part of the animal kingdom, such as mammals (including humans), birds, reptiles, arthropods, and has even been detected in amoebae [1,2]. Wild and domestic ruminants are considered the main reservoir, while small ruminants (goats and sheep) are frequently implicated in human outbreaks and are the target of hygiene and biosecurity measures [3].

In animals, Q fever is described as a subclinical disease (infection is mainly respiratory, although a digestive route has also been described), but when it manifests clinically, it can lead to reproductive disorders characterized by abortions (in the latter part of gestation), stillbirths, and infertility [4].

Infected ruminants excrete Coxiella through various biological fluids such as milk, feces, urine, and especially birth/abort products, which can contain large numbers of microorganisms (up to 10^9^ bacteria/g placenta) [1].

Q fever is a public health and animal health concern because it has an economic impact on livestock and is considered a zoonotic disease. The economic impact derives from the abortions (rate ranging from 3% to 80%) and the measures that must be taken when the disease is transmitted to humans. For example, during the Dutch outbreaks (the largest outbreak ever described, which occurred between 2007 and 2010), the total intervention cost in agriculture was approximately 35,000 euros per disability-adjusted life year (DALY), which included the costs of culling infected animals, breeding prohibition, and vaccination [5]. A recent study found a prevalence of 8.5% in a one-year observational study in which 1212 aborted fetuses were tested with the PCR method [6].

Human cases are often attributed to proximity to, or contact with, infected ruminants (especially small ruminants), as in the outbreaks in the Netherlands, where approximately 4000 cases were reported, and hundreds of chronic forms were diagnosed [7]. In recent years, bovines have been proposed as potential sources of infection for humans, particularly in high-risk areas [8]. This statement is based on the presence of an outbreak associated with the presence of the infection among cattle, as well as the high seroprevalence among predisposed professional figures (cattle farmers and veterinarians) when compared to blood donors in areas with consistent cattle breeding [8,9].

Recently, outbreaks have occurred in several European countries, raising the interest of public health authorities in this emerging zoonotic disease, and calling for the need to implement surveillance plans [10].

Despite this context, information on *Coxiellosis* in Italy is currently limited and based on reports of reproductive problems in livestock farms. These reports are basically insufficient to fully understand the spread and impact of the disease. Circulation has been described in both domestic and wild animals, but large-scale serosurveys have been conducted only in recent years and, to our knowledge, data on regional prevalence are available in only a few regions. In addition, human outbreaks are periodically described (especially in Northern Italy), usually associated with direct or indirect contact with infected herds [11,12]. The high seroprevalence rates (between 5 and 20%) identified in previous studies suggest a wide distribution throughout the peninsula [13,14,15]. No previous observational study has investigated the epidemiological situation of this pathogen in the Campania region, Southern Italy. The aim of this work was to estimate the seroprevalence of Q fever in dairy cows and water buffalo, as well as to identify individual risk factors for *C. burnetii* seropositivity.

## 2. Results

A total of 73 out of 626 dairy bovines tested positive for the presence of specific antibodies to *C. burnetii* (data on sampled animals are available in Appendix A). The overall seroprevalence was 11.7% (CI 95% 9.15–14.2), distributed among the provinces as follows (the spatial distribution of positive farms is shown in Figure 1): Avellino 13% (CI 95% 6.4–19.5), Benevento 8% (CI 95% 2.7–13.3), Caserta 12.4% (CI 95% 7.8–17.1) and Salerno 12% (CI 95% 7.8–16.2). The highest apparent prevalence at the animal level was found in Avellino province, while the lowest prevalence was observed in Benevento province, but without statistical significance in both cases (Chi-square test: DF = 4, *n* = 626, *p* = 0.66). At the farm level, the prevalence was 81.1% (CI 85% 65.8–90.5) and only 7 out of 37 farms had no seropositive animals.

Univariate analysis revealed that some individual factors were associated with *Coxiella* seropositivity (Table 1). Cattle had significantly higher seroprevalence (14.3%; CI 95% 10.9–17.7) than water buffalo (6.5%, CI 95% 3.2–9.8; Chi-square test: DF = 2, *n* = 626, *p* = 0.004). Seroprevalence was found to be higher in animals older than 24 months (6.2% CI 95% 2–10.5; Chi-square test: DF = 2, *n* = 626, *p* = 0.032) and living with other ruminant species (15%, CI 95% 10.8–19.3; Chi-square test: DF = 2, *n* = 626, *p* = 0.021). There were no differences between animals kept stall-fed and partially grazed (Chi-square test: DF = 2, *n* = 626, *p* = 0.2).

Age groups, species, coexistence with other ruminant species, and housing types were thus chosen for the logistic regression model, which revealed that seroprevalence was significantly higher (*p* = 0.036) in cattle than in water buffalo (Table 2), with almost four times the odds of being positive in cattle (OR = 3.95, CI 95% 1.1–14.2). Similarly, we reported significantly higher seroprevalence in adults when compared to young animals (OR = 1.3, CI 95% 1.2–1.4) (Table 2).

## 3. Discussion

Q fever is a widespread infection with outbreaks regularly reported throughout Europe, causing significant economic losses, and increasing the risk of human infection. The collection and interpretation of surveillance data is critical for assessing public health risk and implementing control measures [16,17]. We conducted a serosurvey in the dairy bovine population to confirm the wide distribution of *C. burnetii* in the Campania region, because the epidemiological situation of Q fever in this region has never been defined. The individual seroprevalence was 11.7% (CI 95% 9.15–14.2), which agreed with the results obtained in other Italian regions among ruminant populations. Only two large-scale studies were performed on cattle, with a seroprevalence of 5.28% in Sicily, and 12% in central Italy [14,18]. Similar studies in small ruminant populations revealed a seroprevalence of 15.9% in the Piedmont region and 9.8% in Sardinia [13,15]. There is no data for the buffalo population, even though this disease has been documented as a major cause of abortion in this species [19].

The positive animals were evenly distributed throughout the territory, as evidenced by the similar seroprevalences found in the different provinces (Figure 1). Our data showed that seroprevalence in the province of Avellino appears to be practically unchanged in comparison to what Capuano et al. observed in 2002, when 14.4% of cattle tested positive for the immunofluorescence assay (IFA) [20]. It was reasonable to expect a lower seroprevalence value as a result of improved farm hygiene, because the farms’ biosafety levels at the time (twenty years ago) was clearly lower than that of modern farms. However, the assay used in Capuano et al.’s work was based on *Coxiella burnetii* phase II antigens, whereas modern ELISAs (including the commercial assay used in our study) are based on a combination of phase I and II antigens [21]. Furthermore, the IFA protocol used in Capuano et al.’s work was validated only in humans before being adapted for veterinary use. The seroprevalence found in Avellino province in 2002 may have been underestimated due to the lower sensitivity of this IFA compared to modern ELISA.

Seroprevalence rates in other European bovine populations ranged from 6 to 15% in previous surveys. In Germany, Spain, and Albania, seroprevalence rates were 7.8%, 6.7%, and 7.9%, respectively [22,23,24]. Similarly, a large-scale study in Northern Ireland of 5182 animals found a seroprevalence of 6.2% [25]. A serosurvey recently conducted in Poland revealed a seroprevalence of 4.18% [26]. In the Netherlands, just before the 2007–2011 epidemic, the highest level ever reported in apparently healthy animals (16%) was described [27]. Overall, these findings indicate that *C. burnetii* is common in bovine populations, though small differences could be due to differences in epidemiologic situations, as well as the characteristics of the ELISA kit used and the sample size.

In the buffalo population, the epidemiological background is slightly different. To the best of our knowledge, no European countries have data on this species, and only a few countries, including India, Egypt, and Thailand, have conducted large-scale buffalo serosurveys, with seroprevalence rates ranging from 5 to 20% [28,29,30,31,32]. In our study, the water buffalo population had lower seroprevalences in comparison to the cattle population. Prior studies in other countries (India, Egypt, Thailand) where buffalo breeding is widespread, confirmed this information, which was also supported by molecular tools [28,29,30]. This differs with the findings of an Indian study, which found a seroprevalence of 28% in buffalo and 13.6% in cattle with no apparent reason to explain this result, excluding the different epidemiological situations, number of samples, and types of test used [29]. Additional research will be required to determine the difference in epidemiology among different ruminant species (strain involved, predominant type of and route of infection, etc.). Ruminants, like humans, have a strong and early reaction (2–3 weeks after infection) against phase II (PhII), followed by an increase in antibodies against phase I after a few weeks. It will be intriguing to investigate the characteristics of immune response in two different species and see if some of them are species-specific [7].

The univariate analysis found several factors correlated to higher seroprevalence, such as age, species, and co-living with other ruminant species. Our findings suggested the absence of correlation (Chi-square test: DF = 2, *n* = 626; *p* = 0.2) between *Coxiella* seropositivity and the type of housing (partially grazed compared to housed animals). Previous studies indicated that animals confined totally indoors have a higher risk of exposure than animals that graze throughout the year, possibly as a result of the confined pathogen excretion by infected animals when bred in close spaces and in intensive systems [28,29,33]. Other research, however, support the idea that pastures pose a concern because of the possibility for contamination by ticks, domestic and wild animals, and problems with putting hygiene precautions into practice [13,20].

In accordance with previous findings, we observed that dairy had a higher risk (Chi-square test: DF = 2, *n* = 626, *p* = 0.021) when bred in the presence of other ruminant species. One possible explanation for this evidence is that some ruminant species (particularly goats and sheep) represent the main reservoir and amplification hosts, as exampled by the higher prevalence of these species [13,32,34]. Although it would be utopian or impossible to eliminate mixed herds, it is critical to remember this statement whenever animals or humans come into contact with these animals.

In the univariate analysis, age was also identified as a risk factor. Other studies have found that the risk of being seropositive increases with age due to longer exposure time [14,28,31,32].

Only species and age were positively correlated with the presence of antibodies against *C. burnetii* in the logistic regression model. The odds of seropositivity were higher in adults (OR: 4.75; 95% CI 2.20–10.26; *p* < 0.001) than in younger animals, as well as the odds were higher in cattle than in buffalo.

Serological evidence in humans, domestic ruminants, exposed workers (veterinarians, farmers, and slaughterhouse workers), pets, wildlife, and zoo animals has been described in Italy, as well as molecular evidence in humans, several mammals, birds, ticks (some of which can transmit the infection to humans), in the environment (urban river water), milk, and milk products [35,36,37,38,39,40,41,42]. Given the relevant presence of Q fever in the Italian eco-systems, disease control strategies would be critical to prevent or better manage potential outbreaks. Our work aims to raise awareness regarding this pathogen by improving the information about the seroprevalence among domestic ruminants, identifying some risk factors and providing serological evidence for a new species (water buffalo). This is the first study to conduct a *Coxiellosis* serosurvey in the Campania region, and it suggests that livestock are contributing to the spread of Q fever in the area, posing a serious threat to human health. Some variables (age, species, and being kept with other susceptible species) have been associated with *Coxiella* exposure, as described in the literature, and this information can be used to improve prevention and control strategies for the implementation of the One Health approach.

Further research is required to assess the pathogen’s role in ruminant abortion in the Campania region, including molecular approaches on different matrices (milk, vaginal fluids, or abortion products), as well as determining the circulating genotypes (for example, using Multiple-Locus Variable Number Tandem Repeat Analysis),which would allow identifying the genotypes involved in human and animal outbreaks (as has already been done in central Italy) and comparing genotypes across different regions and nations [35].

## 4. Materials and Methods

### 4.1. Study Area and Sampling

The present study was conducted in Campania (41°00000″ N–14°30000″ E), a region in Southern Italy (with an area of 1,359,000 ha) bordering the Mediterranean Sea (coastline of 350 km). In this area, a total of 690.000 ruminants are raised, including 165,000 cattle and 300,000 water buffalo (Banca Dati Nazionale dell’ Anagrafe Zootecnica, https://www.vetinfo.it/j6_statistiche/, accessed on 1 April 2022).

Given the absence of similar studies in the Campania region, we decided to assume an expected prevalence of 0.5 (i.e., 50%), an absolute precision of 5%, and a 95% confidence interval.

The sample size was calculated by Thrusfield’s formula as follows:$$n=\frac{Z^{2}\times P\left(1-P\right)}{d^{2}}$$
where: *Z* = 1.96 for a confidence level of 95%, P = expected prevalence, *d* = accepted error, *n* = sample size. This approach was conservative, and this prevalence value yielded the largest sample size for the same absolute precision [43]. Sampling began in October 2020 and coincided with blood collection performed by state veterinarians for the national Brucellosis eradication program (approval from the ethics committee was not required for this study). A total of 21 cattle farms and 16 water buffalo farms located in four provinces were obtained from the study area in 29 different districts. Only unvaccinated farms were selected, and a total of 412 samples from dairy cows and 214 samples from water buffalos were randomly collected.

### 4.2. Commercial ELISA Assay

Blood samples were collected from the tail vein using a vacutainer. Each sample was then centrifuged at 1000× *g* for 10 min to separate the sera, 2 mL aliquots of which were collected and stored at −20 °C until assayed. Each serum sample was tested for *C. burnetii* specific IgG antibodies using the “Q Fever (*C. burnetti*) Antibody” ELISA Test Kit (IDEXX Laboratories), according to the manufacturer’s instructions. This test consists of an ELISA coated with native antigens (extracted from Phase I and Phase II Nine Mile strains). This assay is widely used for routine diagnostics and large-scale epidemiological studies. To highlight the spatial distribution of positive animals, a map was created using the geographic information system (GIS) QGIS 3.22 (QGIS Association. http://www.qgis.org, accessed on 1 April 2022).

### 4.3. Statistical Analysis

Prevalence at animal level was calculated by dividing the number of positive bovines by the total number of bovines screened. The information collected during the sampling was used for risk factor analysis. First, univariate analysis at the animal level using chi-square statistics was used to evaluate risk factors for Q fever positivity (expressed as binary variables). A *p*-value less than 0.05 was considered significant. Before being included in the multivariable logistic regression analysis, variables associated with *p* < 0.2 in the univariable screening were checked for multicollinearity. After checking for collinearity, a logistic regression model was used to assess the relationship between each variable and the prevalence of *Coxiella* antibodies: outcome (positive or negative) represented the dependent variable, age (expressed in years) represented the continuous variable, and the other variables (type of housing, coexistence with other ruminants and province) were the predictors.

The odds ratios (ORs) and their 95 percent confidence intervals were used to calculate the degrees of association between independent variables and Q fever serological status (CI). In the final model, *p* < 0.05 was considered significant. Statistical analysis was performed with MedCalc Statistical Software version 16.4.3 (MedCalc Software, Ostend, Belgium; www.medcalc.org, accessed on 1 April 2022) and JMP version 14.1.0 (SAS Institute Inc., Cary, NC, USA).

## 5. Conclusions

The individual and herd level seroprevalence findings in the present study suggest that Q fever is widespread among dairy cattle and water buffalo in the study area, calling for the need to implement a surveillance strategy for this zoonotic disease. According to the literature, some variables (age, species, and keeping with other susceptible species) have been associated with *Coxiella* exposure. This data needs to be improved by identifying other risk factors involved in disease transmission, particularly those addressing target measures to reduce the risk of transmission among herds, so that it can be used to improve prevention and control strategies for the implementation of the One Health approach. In general, further research is required to establish the role of *Coxiella burnetii* as an abortion pathogen in ruminants and as a zoonotic agent in the Campania region.

## Figures and Tables

**Figure 1 pathogens-11-00901-f001:**
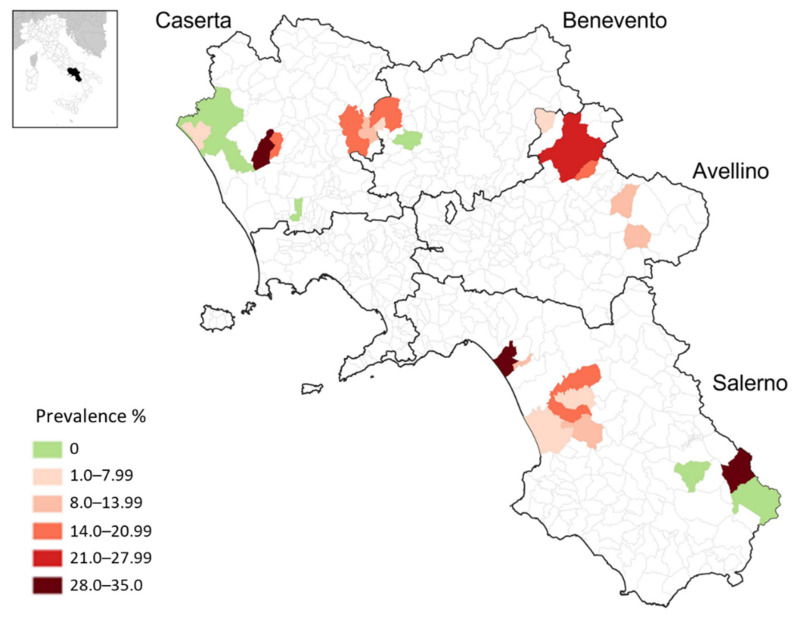
A map of the Campania Region (Italy) showing the spatial distribution of seropositive dairy bovines for Coxiella burnetii.

**Table 1 pathogens-11-00901-t001:** Seroprevalence of Q fever and univariate analysis (chi-square) of potential risk factors for Q fever seropositivity.

Variable and Level	*n*	Positive	%	95% CI	χ^2^	*p*
Total	626	73	11.7	9.15–14.2		
Species						
Cattle	412	59	14.3	10.9–17.7		
					8.27	0.004
Buffalo	214	14	6.5	3.2–9.8		
Province						
Avellino	100	13	13	6.4–19.5		
Benevento	100	8	8	2.7–13.3	1.62	0.66
Salerno	233	28	12	7.8–16.2		
Caserta	193	24	12.4	7.8–17.1		
Age						
≤24 months	128	8	6.2	2–10.5		
					4.57	0.032
>24 months	498	65	13	10.1–16		
Housing						
Partly grazed	120	10	8.3	3.4–13.3		
					1.57	0.2
Stallfed	506	63	12.4	9.6–15.3		
Coexistence with other ruminant species						
Yes	273	41	15	10.8–19.3		
					5.3	0.021
No	353	32	9	6.1–12.1		

**Table 2 pathogens-11-00901-t002:** Logistic regression model for the association of potential risk factors with Q fever seropositivity. Only variables with significant categories (*p* < 0.05) are shown, results on non-significant factors are available in Appendix A.

Variable	Exp (B)	SE	OR	CI OR%	*p*-Value
Species (cattle)	1.37	0.65	3.95	1.1–14.2	0.036
Age	0.25	0.05	1.3	1.2–1.4	<0.001

## Data Availability

Not applicable.

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
