# Peer review of "Serological Evidence of Q Fever among Dairy Cattle and Buffalo Populations in the Campania Region, Italy"

_pathogens, 2022, doi:10.3390/pathogens11080901_

Round 1

Reviewer 1 Report

The work of Ferrara and co-workers aimed to describe the seroprevalence of Q fever in dairy cows and water buffalos in the Campania region in Southern Italy using a commercial ELISA kit. The authors found an overall seroprevalence of 11.7% of C. burnetii in this region. Several factors were positively associated with higher seroprevalence, such as species, age, and coexistence with other ruminant species. The findings show a seroprevalence similar to that observed in previous studies in other Italian regions and European countries and suspect a widespread presence of C. burnetti in Campania. Since zoonotic infections are usually due to contact with infected ruminants, the authors recommend that surveillance for Q-fever needs to be improved in affected regions.

The study presented is technically relatively simple, with descriptive content but accurately performed. There are no significant complaints, but it would have been very interesting for the reader if the authors could have provided information on the C. burnetti genotype(s) involved in the infections in Campania. Although the work aimed to estimate the seroprevalence of Q fever and to identify individual risk factors for seropositivity, the authors should still be able to include genotyping analyses (e.g., using the MLVA/VNTR typing method) in their submitted study.

Moreover, in addition to seroprevalence, it would also be interesting to know whether antibodies against phase I or II coxiella were detected in cows and water buffalos. This distinction by commercial phase-specific ELISAs could provide helpful information on whether early/acute and/or late/chronic stages of infections are present in the two investigated species.

These points could still be addressed in the discussion for future studies.

Author Response

We thank the reviewer for his/her valuable comments. We have added some information about the topics indicated by the reviewer (Line 158-160 and 190-191) and we have added points to be addressed in future studies (genotyping analyses using the MLVA/VNTR typing method and differentiation of type of infection in the two investigated species).

Reviewer 2 Report

The manuscript "Serological evidence of Q fever among dairy cattle and buffalo populations in the Campania region, Italy" provides valuable information about an important zoonotic disease. Though it analyses data of exclusively one region, it can count on the interest of a wider scientific audience because it contributes to the general knowledge of Q fever's epidemiology (especially in water buffalo).

After mild modifications, the manuscript is suggested for publication.

- The details of data analysis are advised to provide. In MatMet section, in the statistical analysis subsection, the authors claimed that age was considered a continuous variable, which was expressed in years. This contradicts Table 1 where age was expressed as above and under 24 months. This suggests that age was used as a binary variable.

- In Table 2, data on non-significant factors should be given. These data might provide some additional information. Further analysis of the "coexistence with other ru spec" variable seems promising.

- In the Conclusion section, the main messages of the manuscript deserve some words. Not just considerable seroprevalence but age, species, and common keeping with other susceptible species should be mentioned.

- The sentence in Line 63-64 promises a deeper analysis of the influence of different risk factors. In the Discussion section, this analysis can be presented with the application of the One Health approach.

Minor notes:

Line 45 - mistyping

Line 53 - "4.000" should be replaced with 4000 or 4,000

Line 58 - "areas populated in great part by bovines" sounds strange

Line 102-103 (Table2) - "stabulation" should be replaced with "housing" or another common term

Line 149-152 - the sentence is hardly understandable

Line 156 - "recovery" should be replaced with "housing"

Line 267, 270-272, 295-296 - the form of the references should be checked 

Author Response

After mild modifications, the manuscript is suggested for publication.

1) The details of data analysis are advised to provide. In MatMet section, in the statistical analysis subsection, the authors claimed that age was considered a continuous variable, which was expressed in years. This contradicts Table 1 where age was expressed as above and under 24 months. This suggests that age was used as a binary variable.

Response: We have provided the details of the data analysis in Table S1. We specifiy that age was used as a binary variable (line 229) in the univariate analysis and as a continuous variable in the logistic regression model.

2) In Table 2, data on non-significant factors should be given. These data might provide some additional information. Further analysis of the "coexistence with other ru spec" variable seems promising.

Response: Information on non-significant factors has been provided in a Table S2 (supplemental file).

3)In the Conclusion section, the main messages of the manuscript deserve some words. Not just considerable seroprevalence but agespecies, and common keeping with other susceptible species should be mentioned.

Response: Information has been provided (Line 246-249).

4)The sentence in Line 63-64 promises a deeper analysis of the influence of different risk factors. In the Discussion section, this analysis can be presented with the application of the One Health approach.

Response: Information has been provided (Line 185-188).

Minor notes:

Line 45 – mistyping

Response: Corrected

Line 53 - "4.000" should be replaced with 4000 or 4,000

Response:”4.000” has been replaced by 4,000 (Line 53).

Line 58 - "areas populated in great part by bovines" sounds strange

Response: This sentence has been reformulated: “This statement is based on the presence of an outbreak associated with the presence of the infection among cattle, as well as the high seroprevalence among predisposed professional figures (cattle farmers and veterinarians) when compared to blood donors in areas with consistent cattle breeding” (Line 58)

Line 102-103 (Table2) - "stabulation" should be replaced with "housing" or another common term

Response: “Stabulation” has been replaced by “housing” as suggested by reviewer.

Line 149-152 - the sentence is hardly understandable

Response: The sentence has been rephrased as follows: “This differs with the findings of an Indian study, which found a seroprevalence of 28 % in buffalo and 13.6 % in cattle with no apparent reason to explain this result, excluding the different epidemiological situation, number of samples, and type of test used” (Line 152-154).

Line 156 - "recovery" should be replaced with "housing"

Response: “Recovery” has been replaced by “housing” as suggested by the reviewer.

Line 267, 270-272, 295-296 - the form of the references should be checked 

Response: The form of reference has been checked.

Reviewer 3 Report

It is very nice manuscript about  seroprevalence of Q fever. I have only a few remarks.

Introduction line 72-73 - in the amin of the studies lack is buffalo there is only estimate the seroprevalence of Q fever in dairy cows.

Discussion line 136-137 - reference about serosurvey in Poland is not recent (from 2014) there are avilable more current publications from Poland.

Figure 1 - I suggest add the names of the region in the map.

Author Response

Introduction line 72-73 - in the amin of the studies lack is buffalo there is only estimate the seroprevalence of Q fever in dairy cows.

Response: The aim has been rephrased as follows: "The aim of this work was to estimate the seroprevalence of Q fever in dairy cows and water buffalo as well as to identify individual risk factors for C. burnetii seropositivity". (Line 73-74)

Discussion line 136-137 - reference about serosurvey in Poland is not recent (from 2014) there are avilable more current publications from Poland.

Response: A more recent reference concerning the seroprevalence in Poland has been provided. (Line 140)

Figure 1 - I suggest add the names of the region in the map.

Response: As suggested by the reviewer, the names of the provinces have been provided on the map.
